# Iron-catalyzed radical Markovnikov hydrohalogenation and hydroazidation of alkenes

Jonas Elfert[1], Nils Lennart Frye[1], Isabel Rempel[1], Constantin Gabriel Daniliuc [1] & Armido Studer [1] ✉

We herein report radical hydroazidation and hydrohalogenation of mono-, di- and trisubstituted alkenes through iron catalysis. The alkene moiety that often occurs as a functionality in natural products is readily transformed into useful building blocks through this approach. Commercially available tosylates and α-halogenated esters are used as radical trapping reagents in combination with silanes as reductants. The reported radical Markovnikov hydroazidation, hydrobromination, hydrochlorination, and hydroiodination occur under mild conditions. These hydrofunctionalizations are valuable and practical alternatives to ionic hydrohalogenations with the corresponding mineral acids that have to be run under harsher acidic conditions, which diminishes the functional group tolerance. Good to excellent diastereoselectivities can be obtained for the hydrofunctionalization of cyclic alkenes.

Radical alkene hydrofunctionalization is generally conducted under mild conditions and various functional groups can be introduced with excellent regioselectivity through such a strategy. Cobalt, manganese, and iron complexes have been used as catalysts along these lines[1]. In his pioneering work on radical alkene hydration, Mukaiyama applied cobalt catalysis with $O_2$ as an oxygenation reagent[2]. The Carreira group later developed cobalt-catalyzed alkene hydroazidation and hydrochlorination using tosyl azides and tosyl chlorides as radical traps (Fig. 1A)[3-5]. Recently, this protocol was further optimized by the Zhu group and applied to the modification of polymers[6]. Hydrobromination and hydroiodination have been achieved by Herzon and Ma with $Co(acac)_2$ as a stoichiometric mediator and tosyl bromide and diiodomethane as halogen atom transfer reagents[7]. Very recently, the Ohmiya group published a valuable method for radical alkene hydrohalogenation (F, Cl, Br, and I) using cobalt and iridium dual catalysis[8]. Through a related approach, Lin et al. achieved photocatalytic Markovnikov hydrofluorination with triethylamine trihydrofluoride as F-source[9]. It is important to highlight that these radical hydrohalogenations occur under mild non-acidic conditions and as compared to the classical Markovnikov hydrohalogenation with the corresponding mineral acid, functional group tolerance is significantly higher. Moreover, cationic rearrangements that often occur as side

reactions in the ionic hydrohalogenation are not a problem using the radical approach.

Considering iron-based transformations that are undoubtedly even more attractive, as iron is the most abundant and least toxic metal, current hydrofunctionalization protocols mostly require stoichiometric amounts of iron[10-15]. For example, the Boger group developed alkene hydrofunctionalizations using superstoichiometric amounts of iron salt (Fig. 1B)[12,14]. Ishibashi et al. achieved Fe-mediated reductive cyclization of dienes with concomitant bromination or iodination[13]. These Fe-mediated hydrofunctionalizations proceed through the initial formation of a Fe(III) hydride species by reaction of a Fe(III)X complex with a reductant (borohydride or silane) mostly in the presence of alcohol. After metal-hydride hydrogen atom transfer (MHAT) to the alkene, a carbon-centered radical is formed, which can further react with a radical trapping reagent either through addition to a π-system or halogen abstraction to form the targeted product along with a Fe(II)-complex. To achieve Fe-catalysis, the radical adduct or byproduct formed after trapping must be able to reoxidize the Fe(II)-species. An alternative would be the use of a stoichiometric external oxidant, as realized for Co-systems[7,16-18]. Oxidation and regeneration of the Fe(III)−H complex (rate determining step) under reducing conditions pose challenges and inefficient oxidation leads to a buildup of inactive Fe(II)-species that eventually stops catalysis. In fact,

---

[1]Organisch-Chemisches Institut, Universität Münster, Münster, Germany. ✉e-mail: studer@uni-muenster.de

**Fig. 1 | Radical hydrofunctionalizations. A** Co-catalyzed alkene hydro-functionalization. **B** Hydrofunctionalization of alkenes by using super-stoichiometric amounts of an iron salt as a mediator. **C** Reductive radical alkene/alkene cross-coupling applying Fe-catalysis and suggested mechanism for this radical cascade reaction with an α-ester radical as key intermediate to reoxidize the Fe(II)-complex. Reduction of the Fe(III)OEt with a silane to regenerate the Fe(III) hydride is the rate-determining step. **D** General reaction scheme of the herein-introduced Fe-catalyzed alkene hydrohalogenation. Design of the radical X-group transfer reagent.

only a limited number of successful Fe-catalyzed radical hydro-functionalizations have been reported to date. For example, the Boger group realized alkene hydration with air as the oxidant using Fe-catalysis[12]. We developed an iron-catalyzed Mukaiyama-type hydration that utilizes nitroarenes as radical traps[19,20]. Moreover, additional Fe-catalyzed hydrofunctionalizations, like C–C couplings[21–23], hydronitrosation[24], hydroamination[25,26], hydrofluorination[27], and hydroalkynylation[28] have been reported. Further advancements have recently been made by the groups of Shenvi and Baran, who achieved coupling reactions using catalytic amounts of iron porphyrin complexes[29,30]. These complexes are able to form an intermediate alkyl iron complex that can engage in coupling reactions with alkyl radicals through homolytic substitution.

The Baran group established Fe-catalyzed reductive cross-coupling of electron-rich alkenes with α,β-unsaturated esters or ketones (Fig. 1C)[31]. In these couplings, the initial MHAT occurs at the electron-rich alkene, and the H-adduct C-radical further reacts with the Giese-acceptor to give an α-keto or an α-ester C-radical that are both able to reoxidize the Fe(II)-complex. The critical Fe(II)-oxidation step was studied in more detail by the groups of Holland and Poli who suggested that the alcohol is of importance, as oxidation likely proceeds through a concerted proton-coupled electron transfer (CPET) from a Fe(II)/alcohol complex to the C-radical[32]. In Fe-catalyzed hydrofunctionalizations where catalytic turn-over without additional oxidant is achieved, an α-keto or an α-ester C-radical often appear as key intermediates[22,26,33,34].

In this work, we present iron-catalyzed radical alkene hydro-functionalizations, leveraging the oxidative capabilities of α-ester C-radicals to facilitate catalytic turnover of the redox cycle (Fig. 1D). The hydrofunctionalization reagents were designed according to the fol-lowing criteria. The X-atom transfer reagent used as the radical trap should (a) react efficiently with secondary and tertiary C-radicals, and (b) after successful X-atom transfer generates an α-ester radical that is reducible by a Fe(II)-complex. Based on these requirements, we selected readily available α-halo esters, where depending on the α-substituents,

X-atom-transfer to nucleophilic C-radicals are known to be efficient (at least for Br and I)[35]. To our knowledge, Fe-catalyzed radical Markovnikov hydrohalogenation has not been reported to date.

## Results

### Hydrobromination and hydroiodination−reaction optimization

We commenced our studies by using the terminal alkene **1a** as the model substrate in combination with commercial α-bromoisobutyric ester **2a** as the Br-transfer reagent. Pleasingly, with 1.5 equivalents of **2a**, Fe(dpm)₃ (Hdpm = dipivaloylmethane) as catalyst (10 mol%) in combination with the RubenSilane (PhSiOiPrH₂)[36] (2 equiv.) in a THF/iPrOH solvent mixture, the targeted secondary bromide **3a** was obtained in 85% yield (Table 1, entry 1). Replacing the silane **4** with commercial phenylsilane led to a slight decrease in yield (entry 2). A lower conversion and lower yield were noted with Fe(acac)₃ (entry 3) and a further diminished yield was achieved with Fe(acac)₃ in combi-nation with phenylsilane as the hydride source (56%, entry 4). How-ever, a better result was noted for the Fe(acac)₃/PhSiH₃ couple by using methanol as solvent (76%, entry 5). Upon decreasing the amount of phenylsilane (1 equiv.) and prolonging the reaction time to 2 days we could further improve the yield to 84% (entry 6). Increasing the tem-perature to 40 °C allowed to shorten the reaction time to 24 h accompanied by a slight loss in yield (entry 7). The best result was achieved by decreasing the amount of radical trap to 1.2 equivalents, leading to an excellent 98% yield of **3a** (entry 8).

Unfortunately, when we applied these optimized conditions to the hydrobromination of the 1,1-disubstituted alkene **1b**, which reacts through a tertiary alkyl radical, only a 9% yield of **3b** was obtained (entry 9). An improved result was noted for the hydrobromination of **1b** upon switching to Fe(dpm)₃ as precatalyst in combination with the RubenSilane (2 equiv.) in THF/iPrOH (27%, entry 10). The conversion was further improved by lowering the amount of the trapping reagent **2a** (1.1 equiv.) and **3b** was formed in 79% yield (entry 11). The best result was achieved by lowering the amount of silane **4** to 1.2 equivalents

**Table 1 | Optimization of the Fe-catalyzed radical Markovnikov hydrobromination and hydroiodination with various X-transfer reagents using alkenes 1a and 1b as model substrates**

| Entry[a] | Fe–H source | Solvent | Radical trap | Conversion | Yield |
|---|---|---|---|---|---|
| 1 | Fe(dpm)₃, **4** (2.0 eq.) | THF/iPrOH | **2a** (1.5 eq.) | 98% (**1a**) | 85% (**3a**) |
| 2 | Fe(dpm)₃, PhSiH₃ (2.0 eq.) | THF/iPrOH | **2a** (1.5 eq.) | 99% (**1a**) | 79% (**3a**) |
| 3 | Fe(acac)₃, **4** (2.0 eq.) | THF/iPrOH | **2a** (1.5 eq.) | 90% (**1a**) | 79% (**3a**) |
| 4 | Fe(acac)₃, PhSiH₃ (2.0 eq.) | THF/iPrOH | **2a** (1.5 eq.) | 93% (**1a**) | 56% (**3a**) |
| 5 | Fe(acac)₃, PhSiH₃ (2.0 eq.) | MeOH | **2a** (1.5 eq.) | 98% (**1a**) | 76% (**3a**) |
| 6[b] | Fe(acac)₃, PhSiH₃ (1.0 eq.) | MeOH | **2a** (1.5 eq.) | 94% (**1a**) | 84% (**3a**) |
| 7[c] | Fe(acac)₃, PhSiH₃ (1.0 eq.) | MeOH | **2a** (1.5 eq.) | 96% (**1a**) | 81% (**3a**) |
| 8[b] | Fe(acac)₃, PhSiH₃ (1.0 eq.) | MeOH | **2a** (1.2 eq.) | 98% (**1a**) | 98% (**3a**) |
| 9 | Fe(acac)₃, PhSiH₃ (1.5 eq.) | MeOH | **2a** (1.5 eq.) | 41% (**1b**) | 9% (**3b**) |
| 10 | Fe(dpm)₃, **4** (2.0 eq.) | THF/iPrOH | **2a** (2.0 eq.) | 57% (**1b**) | 27% (**3b**) |
| 11[b] | Fe(dpm)₃, **4** (2.0 eq.) | THF/iPrOH | **2a** (1.1 eq.) | 89% (**1b**) | 79% (**3b**) |
| 12[b] | Fe(dpm)₃, **4** (1.2 eq.) | THF/iPrOH | **2a** (1.1 eq.) | 90% (**1b**) | 81% (**3b**) |
| 13 | Fe(acac)₃, PhSiH₃ (2.0 eq.) | MeOH | **2b** (3.0 eq.) | 33% (**1a**) | 12% (**5a**) |
| 14 | Fe(acac)₃, PhSiH₃ (2.0 eq.) | MeOH | **2c** (3.0 eq.) | n.d. (**1a**) | 8% (**5a**) |
| 15 | Fe(dpm)₃, **4** (2.0 eq.) | THF/iPrOH | **2d** (3.0 eq.) | 49% (**1a**) | 0% (**5a**) |
| 16 | Fe(acac)₃, PhSiH₃ (2.0 eq.) | MeOH | **2e** (3.0 eq.) | 33% (**1a**) | 25% (**5a**) |
| 17 | Fe(acac)₃, PhSiH₃ (2.0 eq.) | MeOH | **2f** (3.0 eq.) | 0% (**1a**) | 0% (**5a**) |
| 18[d] | Fe(acac)₃, PhSiH₃ (1.0 eq.) | MeOH | **2e** (1.0 eq.) | 87% (**1a**) | 84% (**5a**) |
| 19[e] | Fe(acac)₃, PhSiH₃ (1.0 eq.) | MeOH | **2e** (1.0 eq.) | 81% (**1a**) | 81% (**5a**) |

[a]Reactions were carried out on a 0.1 mmol scale. The reactions were stirred until no further conversion was observed by GC analysis. Conversion and yield were determined by GC analysis using dodecane as an internal standard.
[b]Stirred for 48 h at rt.
[c]Reaction run at 40 °C.
[d]For 4 days on a 0.5 mmol scale.
[e]At 40 °C for 30 h.

(81%, entry 12). Continuously extending the reaction time failed to yield improved results. Instead, it resulted in stagnant conversion.

Next, we optimized the hydroiodination of the terminal alkene **1b**. With the α-iodoisobutyric ester **2b** as the I-transfer reagent (3 equiv.) under the above-optimized hydrobromination conditions, the secondary alkyl iodide **5a** was formed in low yield (12%, entry 13). We therefore tested other α-iodo esters as trapping reagents and with the ester **2c** yield further decreased to 8% (entry 14). With the α,α-difluoroester **2d** as a radical trap, product **5a** was not formed (entry 15). Pleasingly, the α-iodopropionic ester **2e** gave **5a** in an improved yield (25%), but conversion remained low (entry 16) and no reaction was noted with the α-phenyl-α-iodo ester **2f** as the radical trap (entry 17). Optimization studies were therefore continued with the most promising reagent **2e**. Conversion and yield could be significantly increased by reducing the amounts of phenylsilane and radical trap to just one equivalent each (entry 18). Under these conditions, the reaction took longer and after 4 days, an 89% yield was achieved. Unfortunately, prolonging the reaction time did not result in additional conversion, as the reaction stopped. Upon running the hydroiodination at 40 °C, reaction time could be shortened to 30 h at a slight expense in yield (entry 19). For a full optimization study, please refer to the Supplementary Material. Unfortunately, as tertiary alkyl iodides are not stable under the reaction conditions, hydroiodination of the 1,1-disubstituted alkene **1b** was not successful.

**Hydrochlorination and hydroazidation−reaction optimization**
We next addressed the hydrochlorination of the alkenes **1a** and **1b**. Encouraged by the successful hydrobromination with **2a**, we first selected 2-chloroisobutyricacid methyl ester as the radical chlorination reagent. However, under all tested conditions the Fe-catalyzed radical hydrochlorination failed for these alkenes. This is not unexpected as Cl-atom transfer reactions are several orders of magnitude slower than the corresponding bromine atom transfers[37]. We, therefore, switched to commercial tosyl chloride as the Cl-donor that was successfully used before in Fe-mediated (superstoichiometric) Markovinkov hydrochlorination reactions[12] and also in Co-catalyzed alkene hydrofunctionalizations[4]. Thus, secondary and tertiary C-radicals are known to react with tosyl chloride and the challenge lay in finding conditions, where the tosylsulfonyl radical generated after Cl-atom transfer efficiently oxidizes the Fe(II)-complex to close the redox cycle.

Initial optimization studies were conducted on the 1,1-disubstituted alkene **1b** to give the tertiary chloride **7b** (for a full optimization study, please refer to the Supplementary Material). By using Fe(dpm)₃ as precatalyst (10 mol%) in combination with the RubenSilane **4** (2.0 equiv.) and 1.5 equivalents of tosyl chloride **6a**, the desired hydrofunctionalization product **7b** was obtained in 74% yield (Table 2, entry 1). Replacing **4** by phenylsilane afforded a worse result (entry 2) and Fe(acac)₃ in combination with **4** provided a significantly diminished yield (entry 3). Increasing the amount of tosyl chloride to 3

**Table 2 | Optimization of the Fe-catalyzed hydrofunctionalization with commercial tosyl chloride and tosyl azide as the atom or group transfer reagents**

**1a** (R$^1$ = H, R$^2$ = Ph)
**1b** (R$^1$ = Me, R$^2$ = OBz)

**6a** (X = Cl, 1.5 eq.)
**6b** (X = N$_3$, 1.5 eq.)

iron precat. (10 mol%)
silane (2.0 eq.)
───────────────────
THF/iPrOH (1/1)
rt, 24 h

**7a** (R$^1$ = H, R$^2$ = Ph, X = Cl)
**7b** (R$^1$ = Me, R$^2$ = OBz, X = Cl)
**8a** (R$^1$ = H, R$^2$ = Ph, X = N$_3$)
**8b** (R$^1$ = Me, R$^2$ = OBz, X = N$_3$)

| Entry[a] | Fe–H source | Radical trap | Conversion | Yield |
|---|---|---|---|---|
| 1 | Fe(dpm)$_3$, **4** (2.0 eq.) | **6a** (1.5 eq.) | 76% (**1b**) | 74% (**7b**) |
| 2 | Fe(dpm)$_3$, PhSiH$_3$ (2.0 eq.) | **6a** (1.5 eq.) | 64% (**1b**) | 62% (**7b**) |
| 3 | Fe(acac)$_3$, **4** (2.0 eq.) | **6a** (1.5 eq.) | 23% (**1b**) | 23% (**7b**) |
| 4 | Fe(dpm)$_3$, **4** (2.0 eq.) | **6a** (3.0 eq.) | 77% (**1b**) | 77% (**7b**) |
| 5[b] | Fe(dpm)$_3$, **4** (2.0 eq.) | **6a** (1.5 eq.) | 57% (**1b**) | 50% (**7b**) |
| 6 | Fe(dpm)$_3$, **4** (2.0 eq.) | **6a** (1.5 eq.) | 75% (**1a**) | 48% (**7a**) |
| 7 | Fe(dpm)$_3$, **4** (2.0 eq.) | **6a** (1.1 eq.) | 79% (**1a**) | 61% (**7a**) |
| 8[c] | Fe(dpm)$_3$, **4** (3.0 eq.) | **6a** (1.1 eq.) | 90% (**1a**) | 82% (**7a**) |
| 9[d] | Fe(acac)$_3$, PhSiH$_3$ (4.0 eq.) | **6a** (1.5 eq.) | 81% (**1a**) | 80% (**7a**) |
| 10[e] | Fe(acac)$_3$, PhSiH$_3$ (4.0 eq.) | **6a** (1.5 eq.) | 80% (**1a**) | 75% (**7a**) |
| 11 | Fe(dpm)$_3$, **4** (2.0 eq.) | **6b** (1.5 eq.) | 76% (**1a**) | 62% (**8a**) |
| 12 | Fe(dpm)$_3$, **4** (2.0 eq.) | **6b** (1.5 eq.) | 84% (**1b**) | 71% (**8b**) |

[a]Reactions were carried out on a 0.1 mmol scale. The reactions were stirred until no further conversion was observed by GC analysis. Conversion and yield were determined by GC analysis using dodecane as an internal standard.
[b]Connected to air by needle.
[c]Stirred for 72 h.
[d]Stirred for 194 h.
[e]At 40 °C for 48 h.

equivalents did not affect the reaction outcome (entry 4) and exposing the reaction mixture to air had a detrimental effect on the hydrofunctionalization (entry 5). We then switched to the terminal alkene **1a** as the substrate and found that alkyl chloride **7a** was formed in 48% yield under the ideal conditions identified for substrate **1b** (entry 6). The yield could be improved to 61% by lowering the amount of tosyl chloride to 1.1 equivalents (entry 7). We were pleased to find that increasing the amount of silane (3 equiv.) and extending reaction time (72 h) led to a further improvement of the result and **7a** was obtained in 82% yield (entry 8). Notably, similar yields were also achieved with Fe(acac)$_3$ in combination with phenylsilane (4 equiv.). However, the reaction took 8 days to reach high conversion (entry 9). Reaction time could be shortened to 2 days by running the hydrochlorination at 40 °C with little loss in yield (entry 10).

Finally, we also screened conditions for the Fe-catalyzed hydroazidation of the alkenes **1a** and **1b** with commercial tosyl azide **6b** as the group transfer reagent. Of note, Fe-catalyzed hydroazidation is currently unknown to our knowledge. Conditions identified for the hydrochlorination of **1b** turned out to be well-suited for the hydroazidation of the two model alkenes. Thus, the reaction of **1a** with Fe(dpm)$_3$ and the RubenSilane (2 equiv.) in a THF/iPrOH solvent mixture with tosyl azide **6b** (1.5 equiv.) at room temperature for 24 h afforded the alkyl azide **8a** in 62% yield (entry 11). Under the same conditions, the alkene **1b** was successfully converted to the tertiary azide **8b** which was formed in 71% yield (entry 12).

### Hydrofunctionalization of various alkenes–reaction scope

With optimized conditions in hand for the individual hydrofunctionalizations, the reaction scope was examined (Fig. 2). For each monosubstituted alkene, hydrobromination, hydroiodination, hydrochlorination, and also hydroazidation were tested, while for the multisubstituted alkenes hydrobromination, hydrochlorination and hydroazidation were investigated. Considering monosubstituted

alkenes, simple aliphatic systems worked well and regioselective hydrohalogenation, as well as hydroazidation, was achieved in 48–79% isolated yield (**3a**, **5a**, **7a**, and **8a**). Diverse functional groups were tolerated such as a terminal bromide (**3c**, **5c**, **7c**, **8c**, 55–89%), free phenol, and phenol ether (**3d**, **5d**, **7d**, **8d**, 44–74%), as well as a tosylamide with a free NH entity (**3e**, **5e**, **7e**, **8e**, 39–92%). Of note, in all series, the lowest yield was achieved for the hydroazidation. A TBS-protected primary alcohol was tolerated in the hydrobromination (**3f**, 62%) and also in the hydroazidation (**8f**, 40%), while silyl ether decomposition was noted during hydrochlorination and hydroiodination. The epoxide moiety survived the hydrobromination (**3g**, 73%), hydroiodination (**5g**, 43%), and hydroazidation (**8g**, 41%). However, hydrochlorination was not compatible with epoxide functionality. A ketone group was tolerated by all investigated hydrofunctionalizations in good yields (**3h**, **5h**, **7h**, **8h**, 54–81%). Furthermore, 1,2-disubstituted alkenes could also undergo hydrofunctionalizations. For example, *cis/trans*-cyclododecene yielded the hydrobrominated product **3i** in 63% yield. For the corresponding hydroiodination, -chlorination, and -azidation lower yields were achieved (38–48%, **5i**, **7i**, **8i**). For the unsymmetrical 1,2-disubstituted alkene (*Z*)-tricos-9-ene, the products **3j**, **3j′**, **5j**, **5j′**, **7j**, **7j′**, and **8j**, **8j′** could be isolated in 34–67% yields. As expected, we could not separate the two regioisomers and also by NMR they are indistinguishable. As the two alkyl substituents at the double bond are nearly identical, we assume that the regioisomers were formed as a 1:1 mixture, as indicated in the figure.

1,1-Disubstituted alkenes were investigated next. Very good isolated yields were achieved for the hydrobromination, hydrochlorination, and hydroazidation of 3-methylbut-3-en-1-yl benzoate (**3b**, **7b**, **8b**, 74–92%). A phthalimide functionality, as well as a Boc-protected secondary amine, were tolerated and the corresponding tertiary hydrofunctionalization products **3k**, **7k**, **8k**, **3l**, **7l**, and **8l** were obtained in good to very good yields (70–92%). We were pleased to find that

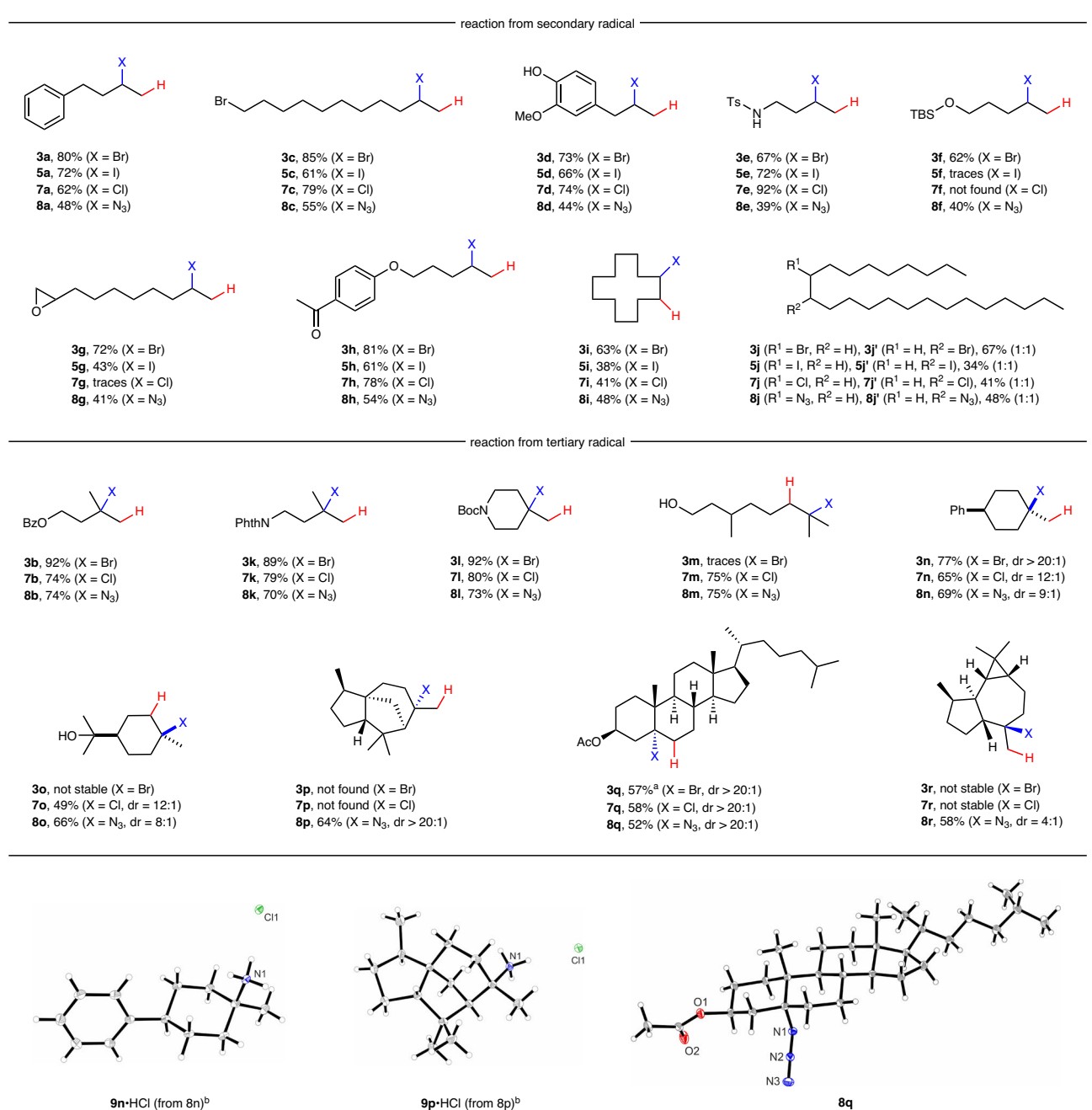

**Fig. 2 | Scope of the Fe-catalyzed hydrohalogenation and hydrofunctionalization of mono-, di- und trisubstituted alkenes including natural products.** X-ray structure of the HCl salts derived from **8n** and **8p**, as well as of azide **8q**. Reactions were carried out on a 0.2–0.5 mmol scale and all yields provided refer to isolated yields. Diastereoselectivity (if applicable) was determined by GC or NMR analysis, and the relative configuration was assigned by X-ray analysis of single crystals or in analogy to a known related compound, see Supplementary Material.[a]NMR yield. [b]See Supplementary Material for the procedure.

trisubstituted alkenes are eligible substrates, as documented by the successful transformation of β-citronellol to provide the tertiary chloride **7m** (75%) and the azide **8m** (75%) that were both formed with complete Markovnikov selectivity. However, only a trace of the product was observed for the corresponding hydrobromination. Unfortunately, hydrofunctionalizations did not work for styrenes, 1,3-dienes, or allylic ethers/alcohols (for an overview of failed substrates, we refer to the Supplementary Material).

We also studied diastereoselective reactions and achieved high *cis*-selectivity for the hydrofunctionalization of (4-methylenecyclohexyl)benzene (**3n**, **7n**, **8n**, 47–77%). (α)-Terpineol could also be hydrochlorinated and -azidated with our three procedures and the hydrofunctionalization products **7o** and **8o** were isolated in 49–66%

yield with similar diastereoselectivity. The hydrobrominated product **3o** was observed by GC with promising selectivity, but could not be isolated due to facile decomposition. With (−)-α-cedrene only hydroazidation was achieved and **8p** was isolated in good yield as a single diastereoisomer. Only very low conversion was noted for the bromination and chlorination, possibly due to the high steric shielding of the tertiary radical. Moreover, hydrochlorination and -azidation of cholesteryl acetate worked well and the corresponding hydrofunctionalization products **7q** and **8q** were obtained in 52–58% yield and perfect diastereoselectivity. The hydrobromination product **3q** could not be isolated, due to decomposition during purification. An NMR yield of 57% could be determined from the crude mixture. (+)-Aromadendrene engaged in the hydroazidation and the tertiary

**Fig. 3 | Mechanistic studies.** Mechanistic experiments (**A**–**C**) and proposed mechanism for the Fe-catalyzed hydrohalogenation of alkenes with α-halo esters (**D**). [a]GC yield.

azide **8r** was isolated in good yield and diastereoselectivity. No clean reaction was achieved for the corresponding hydrohalogenations and isolation of **3r**, as well as of **7r** was not possible due to product instability.

## Mechanistic investigations

To prove whether the ester **2a** indeed serves as a trapping reagent for alkyl radicals, we decomposed dilauroylperoxide (DLP) in hexane at 80 °C in the presence of **2a** and obtained *n*-undecylbromide through a bromine atom transfer reaction (Fig. 3A). The radical nature of the hydrobromination was further supported by a hydrofunctionalization with a concomitant radical 5-*exo*-cyclization. Hence, hydrobromination of the diene **10** gave the cyclization/bromination product **11** as a *cis*:*trans*-mixture of the diastereoisomers in 49% yield (Fig. 3B).

The hydroazidation of natural (−)-α-pinene afforded the expected hydroazidation product **13a** along with the hydroazidation product of limonene **13b** in ~58% combined yield (Fig. 3C). Total conversion of (−)-α-pinene was 64%. **13b** is formed through initial MHAT followed by radical ring-opening to give a tertiary alkyl radical, which can then be

trapped by tosyl azide. Hydrobromination of (−)-α-pinene exclusively leads to the ring-opened product, which is unstable and accompanied by limonene and terpinolene, resulting from the elimination of hydrogen bromide (see Supplementary Material). Taken together, the cyclization product **11** and the ring-opening products observed when employing (−)-α-pinene strongly support the radical nature of the Fe-catalyzed hydrofunctionalization reactions.

The suggested mechanism for the hydrobromination and hydro-iodination with reagents **2a** and **2e** is presented in Fig. 3D. Formation of the Fe−H species is considered to be the rate-determining step in these hydrofunctionalizations[32]. As the RubenSilane is known to be a more efficient reducing reagent for the generation of the metal hydride, it is not surprising that hydrofunctionalizations with **4** are faster as compared to the reactions run with phenylsilane[36]. The Fe−H complex then engages in MHAT to the alkene to give the H-adduct radical **B** along with a Fe(II)-complex. Radical **B** can reversibly trap the Fe(II)-complex to give the corresponding Fe(III)-alkyl complex **A**. In the productive path, alkyl radical **B** is iodinated or brominated by the esters **2a** or **2e** through halogen atom transfer to give the final products **3** or **5**. The

concomitantly generated α-ester radical **C** is then reduced by the Fe(II)/HiOPr through CPET[32] to give ethyl propionate or methyl isobutyrate as byproducts and a Fe(III)OiPr complex that reacts with the silane to regenerate the starting iron hydride complex. The trapping of the alkyl radical with the halo ester should be faster than Fe−H formation, so it is surprising that the reaction time varies depending on the used radical trap and substrate type. Control experiments confirmed that the I-atom transfer with **2e** is faster than the Br-transfer with **2a**, as expected[35]. We currently assume that the longer reaction time for the hydroiodination is due to partial deactivation of the catalyst. For all reactions, conversion slows down as the reaction progresses, sometimes coming to a halt before full conversion is reached. This may be explained by competing unknown side reactions, which might destroy the catalyst or lead to a build-up of a Fe(II)-species that is not reoxidized. We can currently not fully rule out, whether the slower Br-atom transfer from **2a** to the alkyl radical is occasionally mediated by the Fe-catalyst. For the more efficient iodine atom transfer we regard such a scenario as unlikely. Regarding the proposed mechanism for the alkene hydrochlorination and hydroazidation with the tosyl chloride and tosyl azide as radical trapping reagents, please refer to the Supplementary Material.

## Discussion

Radical Markovnikov-type hydrohalogenation of mono-, di- and trisubstituted alkenes has been developed. As halogenation reagents commercially available α-halo esters and tosyl chloride are used. Hydroazidation is possible upon switching to tosyl azide as the trapping reagent under otherwise similar conditions. These highly regioselective radical hydrofunctionalizations are catalyzed with cheap and commercial Fe-catalysts in combination with silane as a reductant. Reactions occur under mild conditions and functional group tolerance is broad. In contrast to the classical Markovnikov hydrohalogenation with the corresponding mineral acids that proceed through cationic intermediates, which are poised for unwanted rearrangements, such a problem does not occur using the radical approach. Moreover, the radical intermediates generated after initial MHAT to the alkene can be harvested to combine the hydrofunctionalization with a typical radical cyclization or fragmentation step. For rigid alkenes, these hydrofunctionalizations occur with excellent diastereoselectivity and complete regioselectivity. Successful hydrofunctionalization of terpenes documents the potential of the herein-introduced processes in synthesis. Importantly, the alkene functionality can be found in many natural products and the introduced functionalities (halides and azide) are valuable entities for follow-up chemical transformation.

## Methods

### Hydrobromination of terminal and 1,2-disubstituted alkenes
Fe(acac)$_3$ (17.7 mg, 50.0 μmol, 10 mol%) was placed in an oven-dried Schlenk tube under argon equipped with a magnetic stir bar and dissolved in dry methanol (4 mL). The alkene (0.50 mmol, 1.0 eq.) and methyl 2-bromo-2-methylpropanoate (78 μL, 0.60 mmol, 1.2 eq.) were added, followed by dropwise addition of phenylsilane (62 μL, 0.50 mmol, 1.0 eq.). The reaction was stirred for 48 h at room temperature. Afterwards, the solvent was evaporated and the residue was purified using flash chromatography to obtain the pure product.

### Hydrobromination of 1,1-disubstituted and trisubstituted alkenes
Fe(dpm)$_3$ (12.2 mg, 20.0 μmol, 10 mol%) was placed in an oven-dried Schlenk tube under argon equipped with a magnetic stir bar and dissolved in dry iPrOH (1 mL) and dry THF (1 mL). The alkene (0.20 mmol, 1.0 eq.) and methyl 2-bromo-2-methylpropanoate (28 μL, 0.21 mmol, 1.1 eq.) were added, followed by dropwise addition of isopropoxy(phenyl)silane (39 μL, 0.22 mmol, 1.0 eq.). The reaction was stirred for 48 h at room temperature. Afterwards, the solvent was evaporated and the residue was purified using flash chromatography to obtain the pure product.

### Hydroiodination of terminal and 1,2-disubstituted alkenes
Fe(acac)$_3$ (17.7 mg, 50.0 μmol, 10 mol%) was placed in an oven-dried Schlenk tube under argon equipped with a magnetic stir bar and dissolved in dry methanol (4 mL). The alkene (0.50 mmol, 1.0 eq.) and ethyl 2-iodo-propanoate (68 μL, 0.50 mmol, 1.0 eq.) were added, followed by dropwise addition of phenylsilane (62 μL, 0.50 mmol, 1.0 eq.). The reaction was stirred for 4 days at room temperature (alternatively 30 h at 40 °C). Afterwards, the solvent was evaporated and the residue was purified using flash chromatography to obtain the pure product.

### Hydrochlorination of terminal and 1,2-disubstituted alkenes
Fe(acac)$_3$ (17.7 mg, 50.0 μmol, 10 mol%) was placed in an oven-dried Schlenk tube under argon equipped with a magnetic stir bar and dissolved in dry iPrOH (2 mL) and dry THF (2 mL). The alkene (0.50 mmol, 1.0 eq.) and pTsCl (143 mg, 0.750 mmol, 1.5 eq.) were added, followed by the dropwise addition of phenylsilane (246 μL, 2.00 mmol, 4.0 eq.). The reaction was stirred for 48 h at 40 °C. Afterwards, the solvent was evaporated and the residue was purified using flash chromatography to obtain the pure product.

### Hydrochlorination of 1,1-disubstituted and trisubstituted alkenes
Fe(dpm)$_3$ (12.2 mg, 20.0 μmol, 10 mol%) was placed in an oven-dried Schlenk tube under argon equipped with a magnetic stir bar and dissolved in dry iPrOH (1 mL) and dry THF (1 mL). The alkene (0.20 mmol, 1.0 eq.) and pTsCl (57.2 mg, 0.300 mmol, 1.5 eq.) were added, followed by dropwise addition of isopropoxy(phenyl)silane (64 μL, 0.40 mmol, 2.0 eq.). The reaction was stirred for 24 h at room temperature. Afterwards, the solvent was evaporated and the residue was purified using flash chromatography to obtain the pure product.

### Hydroazidation of alkenes
Fe(dpm)$_3$ (12.1 mg, 20.0 μmol, 10 mol%) is placed in an oven-dried Schlenk tube under argon equipped with a magnetic stir bar and dissolved in dry iPrOH (1 mL) and dry THF (1 mL). The alkene (0.20 mmol, 1.0 eq.) and pTsN$_3$ (59.2 mg, 0.300 mmol, 1.5 eq.) are added, followed by dropwise addition of isopropoxy(phenyl)silane (64 μL, 0.40 mmol, 2.0 eq.). The reaction is stirred for 24 h at room temperature. Afterwards, the solvent is evaporated and the residue is purified using flash chromatography to obtain the pure product.

## Data availability

Supplementary information and chemical compound information accompany this paper at www.nature.com/ncomms. The data supporting the results of this work are included in this paper or in the Supplementary Information and are also available upon request from the corresponding author.

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

## Acknowledgements

We thank the University of Münster for supporting this work.

## Author contributions

J.E., N.L.F., and A.S. conceived and designed the experiments. J.E., N.L.F., and I.R. performed the experiments and analyzed the data. J.E. and A.S. wrote the manuscript. C.G.D. conducted the X-ray crystal structure analysis.

## Funding

## Competing interests
The authors declare no competing interest.
