## [Peer Review File · Nature Communications]

Iron-Catalyzed Radical Markovnikov Hydrohalogenation and Hydroazidation of AlkenesREVIEWER COMMENTS

Reviewer #1 (Remarks to the Author):

Studer and coworkers report a radical hydrohalogenation and hydroazidation of alkenes through iron-catalyzed metal-hydride hydrogen atom transfer (MHAT) strategy. Secondary and tertiary halides and azides could be generated smoothly when α -halogenated esters and tosylates were used as radical trapping reagents and silanes were used as reductants under mild conditions. Although radical Markovnikov hydrohalogenation and hydroazidation of alkenes have been realized by cobalt-catalyzed or iron-mediated process, the iron catalysis version is yet to be developed until this work. Considering iron is abundant and less toxic, the iron-catalyzed process is attractive for sustainable development and green chemistry. In my opinion, this work is an important progress in the fields of iron catalysis and hydrofunctionalization. I recommend to accept this manuscript for publication in this journal after the following concerns will be addressed:

(1) How about the reactions of 1,2-dialkyl alkenes?

(2) How about the reactions of styrenes?

(3) How about the reactions of 1,3-dienes?

(4) In Figure 2, 8k and 8m refer to azides, while single crystal 8k·HCl and 8m·HCl refer to ammonium salts. These ammonium salts are suggested to be renamed or renumbered to reduce misunderstandings (the readers might consider 8k·HCl and 8m·HCl as hydrochlorides of corresponding azides).

(5) Recent work for catalytic hydrochlorination and hydroazidation of alkenes through MHAT strategy is suggested to be cited: Chem, 2024, 10, DOI: 10.1016/j.chempr.2024.05.021.

Reviewer #2 (Remarks to the Author):

Studer and coworkers reported Fe-catalyzed MHAT functionalization of alkenes. In contrast to Co-catalyzed MHAT functionalization, catalytic process using Fe metal is sometimes not

easy. In this manuscript, the authors carefully selected the radical trapping reagents, which also acted as good oxidants to re-generate Fe(III); thereby enabling Fe-catalyzed Markovnikov hydrohalogenation and hydroazidation. Operationally simple, but carefully optimized conditions gave products in moderate to good yield. The optimized reaction conditions showed reasonably broad scope of alkenes, and good functional group compatibility. It is noteworthy that the reaction proceeded at rt with only silanes and haloesters. Thus, the present method clearly have advantage over previous methods. The reaction mechanism is suitably supported by the mechanistic studies in Figure 3. Thus, I would like to recommend publication of this manuscript after minor revisions.

- 1) As recent Fe-catalyzed reactions, JACS, 2024, 146, 2351 and Science, 2024, 384, 113 by Baran/Shenvi are missing.
- 2) Page 3, typos: “Hdmp = dipivaloylmethan” should be Hdmp = dipivaloylmethane.
- 3) Page 7: “in good to very yields” is strange. Missing word should be added.
- 4) Figure 2: Compound number for 8k-HCl and 8m-HCl (X-ray structures) sounds confusing. As the authors transformed N3 units into amine/salt. The different compound number is suitable, like “9k-HCl derived from 8k”.
- 5) Figure 3: The structure of DLP should be added in Figure 3A.

Reviewer #3 (Remarks to the Author):

Studer and coworkers report a Fe-catalyzed radical markovnikov hydrohalogenation and hydroazidation of alkenes. Tosylates and α -halogenated esters are used as radical trapping reagents in combination with silanes as reductants. The scope and limitations are reasonably well investigated. The mechanism is also reasonable as it is similar to related Fe-catalyzed reactions. However, successful examples of cobalt-catalyzed alkene hydroazidation and hydrochlorination (Angew. Chem. Int. Ed. 2008, 47, 5758–5760) as well as Fe-mediated Markovnikov hydrohalogenation and hydroazidation of alkenes (Org. Lett. 2012, 14, 1428; Org. Lett. 2010, 12, 112) have been reported in existing literature. Therefore, developing the Fe-catalyzed radical markovnikov hydrohalogenation and hydroazidation of

alkenes using the reported similar catalyst system might not be highly impactful. Therefore, I believe this article might be better suited for publication in a more specialized journal rather than Nat. Commun.

Response to the referees' comments

Response to Reviewer #1

Comment 1: Studer and coworkers report a radical hydrohalogenation and hydroazidation of alkenes through iron-catalyzed metal-hydride hydrogen atom transfer (MHAT) strategy. Secondary and tertiary halides and azides could be generated smoothly when α -halogenated esters and tosylates were used as radical trapping reagents and silanes were used as reductants under mild conditions. Although radical Markovnikov hydrohalogenation and hydroazidation of alkenes have been realized by cobalt-catalyzed or iron-mediated process, the iron catalysis version is yet to be developed until this work. Considering iron is abundant and less toxic, the iron-catalyzed process is attractive for sustainable development and green chemistry. In my opinion, this work is an important progress in the fields of iron catalysis and hydrofunctionalization. I recommend to accept this manuscript for publication in this journal after the following concerns will be addressed:

Our response: We appreciate the reviewer's supportive comments.

Comment 2: (1)How about the reactions of 1,2-dialkyl alkenes?

Our response: We have added two 1,2-dialkyl alkenes to the scope study. Hydroazidation and -halogenation can be carried out by the same procedure as for terminal alkenes. The scheme and general procedure have been adapted accordingly. Although we could not separate the two regioisomers (GC) and could also not distinguish them by NMR spectroscopy, it is obvious that the two regioisomers must have been formed as a 1:1 mixture of isomers. We stated that in the text.

Comment 3: (2)How about the reactions of styrenes?

Our response: We have tried the hydrofunctionalizations with styrene and indene and could not obtain any products. We assume that all our radical trapping reagents are not reactive enough to react with the stabilized benzylic radical, as styrenes could be used as substrates in some other hydrofunctionalizations clearly showing that the initial MHAT should work. In case of the iodides, product stability would be another issue. We have commented on limitations in the manuscript and in the supporting information. We have also added an exemplary substrate that shows ketone functional group tolerance (see h-series in Figure 2).

Comment 3: (3)How about the reactions of 1,3-dienes?

Our response: We tested 1,3-dienes, as well as an enyne. Unfortunately, only low conversion for hydroazidation or -bromination was observed with 1 or 2 equivalents of the radical trap. No product could be detected.

Comment 4: (4)In Figure 2, 8k and 8m refer to azides, while single crystal 8k·HCl and 8m·HCl refer to ammonium salts. These ammonium salts are suggested to be renamed or renumbered to reduce

misunderstandings (the readers might considered 8k·HCl and 8m·HCl as hydrochlorides of corresponding azides).

Our response: We agree with the reviewer and have adapted the numbering.

Comment 5: (5)Recent work for catalytic hydrochlorination and hydroazidation of alkenes through MHAT strategy is suggested to be cited: Chem, 2024, 10, DOI: 10.1016/j.chempr.2024.05.021.

Our response: We have added the citation of this recent publication (see yellow marked new reference).

Response to Reviewer #2

Comment 1: Studer and coworkers reported Fe-catalyzed MHAT functionalization of alkenes. In contrast to Co-catalyzed MHAT functionalization, catalytic process using Fe metal is sometimes not easy. In this manuscript, the authors carefully selected the radical trapping reagents, which also acted as good oxidants to re-generate Fe(III); thereby enabling Fe-catalyzed Markovnikov hydrohalogenation and hydroazidation. Operationally simple, but carefully optimized conditions gave products in moderate to good yield. The optimized reaction conditions showed reasonably broad scope of alkenes, and good functional group compatibility. It is noteworthy that the reaction proceeded at rt with only silanes and haloesters. Thus, the present method clearly have advantage over previous methods. The reaction mechanism is suitably supported by the mechanistic studies in Figure 3. Thus, I would like to recommend publication of this manuscript after minor revisions.

Our response: We thank the reviewer for his supportive comments.

Comment 2: 1) As recent Fe-catalyzed reactions, JACS, 2024, 146, 2351 and Science, 2024, 384, 113 by Baran/Shenvi are missing.

Our response: We have addressed these two references in the introduction of the revised manuscript (see text and yellow marked new references).

Comment 3: 2) Page 3, typos: “Hdmp = dipivaloylmethan” should be Hdmp = dipivaloylmethane. 3) Page 7: “in good to very yields” is strange. Missing word should be added.

Our response: Thanks, we have fixed the typos.

Comment 5: 4) Figure 2: Compound number for 8k-HCl and 8m-HCl (X-ray structures) sounds confusing. As the authors transformed N3 units into amine/salt. The different compound number is suitable, like “9k-HCl derived from 8k”.

Our response: we agree and have changed the numbering (see also response to reviewer #1 on the same request).

Comment 6: 5) Figure 3: The structure of DLP should be added in Figure 3A.

Our response: The figure has been changed accordingly.

Response to Reviewer #3

Comment 1: Studer and coworkers report a Fe-catalyzed radical markovnikov hydrohalogenation and hydroazidation of alkenes. Tosylates and α -halogenated esters are used as radical trapping reagents in combination with silanes as reductants. The scope and limitations are reasonably well investigated. The mechanism is also reasonable as it is similar to related Fe-catalyzed reactions.

However, successful examples of cobalt-catalyzed alkene hydroazidation and hydrochlorination (Angew. Chem. Int. Ed. 2008, 47, 5758–5760) as well as Fe-mediated Markovnikov hydrohalogenation and hydroazidation of alkenes (Org. Lett. 2012, 14, 1428; Org. Lett. 2010, 12, 112) have been reported in existing literature. Therefore, developing the Fe-catalyzed radical markovnikov hydrohalogenation and hydroazidation of alkenes using the reported similar catalyst system might not be highly impactful. Therefore, I believe this article might be better suited for publication in a more specialized journal rather than Nat. Commun.

Our response: We regret that reviewer #3 does not see the impact in this work. We believe that iron catalysis is an important field for the development of green reactions, as mentioned by reviewer #1. Compared to other metal-catalyzed hydrofunctionalizations, our approach stands out through the use of mostly commercially available reagents. The catalytic use of metals with low toxicity is also attractive for medicinal chemistry, where metal contamination of final pharmaceuticals is an issue.

REVIEWERS' COMMENTS

Reviewer #1 (Remarks to the Author):

Studer et al. have revised the manuscript by adding more examples and providing more comments for better understandings. I recommend to accept this manuscript for publication in this journal.

A minor point: In page 1 of the revised Supporting Information, the numbers of “References” and “ NMR Spectra” should be 9 and 10, instead of 8 and 9.